# Individual Transferable Quotas (ITQ), Rebuilding Fisheries and Short-Termism: How Biased Reasoning Impacts Management

**Edward J. Garrity**

Richard J. Wehle School of Business, Canisius College, Buffalo, NY 14208, USA; garrity@canisius.edu

**Abstract:** Recent research on global fisheries has reconfirmed a 2006 study that suggested global fisheries would collapse by 2048 if fisheries were not better managed and trends reversed. While many researchers have endorsed rights-based fishery management as a key ingredient for successful management and rebuilding fisheries, in practice the results are mixed and success varies by geographic region. Rights-based approaches such as individual transferable quota (ITQ) provide a necessary help to the important task of rebuilding fisheries, but we assert that they are sometimes less effective due to the human component of the system. Specifically, we examine the issue of setting an appropriate total allowable catch (TAC) in Individual Transferable Quota (ITQ) systems. ITQ are designed on the premise that economic ownership is sufficient incentive to entice fishers to be stewards of the resource. However, an excessive short-term orientation and an affective risk response by fishers can overwhelm feelings of ownership. In such cases, fishers and fishing communities can exert sufficient pressure on TAC setting and reduce the effectiveness of ITQ fisheries toward rebuilding fish stocks. Based on our analysis that draws on cognitive psychology, short-termism, and affective risk, we suggest heightened and wider democratic involvement by stakeholders in co-managed ITQ fisheries along with potential pilot tests of government-assisted financial transfers to help in transitioning ITQ fisheries to sustainable states.

**Keywords:** common pool resources; individual transferable quota; psychological ownership; affective risk; short-termism; systems thinking

## 1. Introduction

Oceans cover over 71% of the Earth's surface and support approximately 3 billion people with food and nutrition. In addition, the oceans are responsible for many important ecosystem services including generating 70% of the atmospheric oxygen necessary for life. Despite these facts, the world's oceans are under assault from climate change, pollution and over-fishing. A 2006 study empirically examined the consequences of ongoing depletion of marine biodiversity, declining fish stocks, reduced water quality, loss of habitat, and less resilient ecosystems [1]. Their study predicted global fisheries collapse by 2048 if fisheries were not better managed and trends reversed. An updated global assessment of 4714 fisheries in 2012 has reconfirmed the original 2006 study and additionally found that 68% of worldwide fisheries have slipped below biomass targets that support maximum sustainable yield (MSY) compared with 63% below targets in 2006 [2]. Climate change is further expected to reduce fishery productivity unless stocks are rebuilt [3].

*The Way Forward: Rights-Based Fishery Management and Multi-Solving*

In cases where rebuilding fisheries has achieved some success, rights-based fishery management was found to be one of many tools necessary to achieve positive outcomes [4,5]. In rebuilding fisheries,

it is generally the case that harvest policies require sharp reductions in fishing effort to allow for rebuilding stocks [2,6,7]. Such actions are often politically infeasible because of the short-term economic sacrifices required, but if these can be managed then long-run gains will outweigh the short-run costs [2,8].

Rights-based fishery management is an attempt to handle environmental exploitation of common pool resources (CPR) by establishing property rights [5,9,10]. The basic idea is to assign property rights so that long-term ecological and conservation goals are aligned with individual economic interests [11–13]. Catch shares, an example incentive-based approach, are allocated privileges to land a portion of the total allowable catch (TAC) in the form of quota shares [13]. Individual transferable quotas (ITQ) are an example of catch shares where the shares are transferable; shareholders have the freedom to buy, sell, and lease quota shares [14]. Since the economic value of quota shares increases when fish stocks are well managed, ITQ shares create an economic incentive for stewardship [15,16].

Although catch shares and incentive-based approaches have potential and theoretical appeal, actual empirical results have been mixed. On the positive side, there is evidence that catch shares eliminate perverse incentives that lead to the "race to fish." Eliminating the "race to fish," which is essentially a competition to land the highest fraction of quota, does lead to additional benefits such as less bycatch and less potential habitat destruction [10,17]. Additionally, it has been shown that catch shares are less likely to crash, or lead to a collapse in harvest [13], they are less likely to have excessive overfishing [18] and they are also found to have better compliance with catch limits [17,18]. However, Melnychuk et al. [18] did not find evidence that catch shares were closer to management targets than other fisheries. These mixed results can be compared with an earlier study by Chu [15], whose survey revealed that in 20 stocks managed by ITQ systems, 12 stocks showed improvements in stock biomass but 8 of the 20 stocks continued to decline after ITQ implementation. Essington et al. [10] studied over 150 fisheries and reported that catch shares tended to dampen variance in fishery landings and exploitation rate but had no effect on population biomass. Finally, some researchers have noted that ITQ systems alone are insufficient to deal with conservation without use of gear restrictions or other ecosystem management tactics [19]. So, despite some of these positive effects, a significant amount of uncertainty remains regarding how well catch shares promote conservation goals and which attributes of catch shares have helped lead to positive outcomes. This does not mean catch shares are ineffective, but rather, catch shares and incentive-based approaches should be complemented by other approaches [20].

Understanding the specific attributes of catch shares that help to promote conservation and fishery goals is important so that management systems can be designed properly to match the characteristics of the environment where they are implemented. Bromley [21] has argued that the conservation benefits of catch shares derive mainly from the use of strict catch quotas (or TAC) to limit fishing mortality. In addition, Essington et al. [10] found that reductions in exploitation rate were strongest in multi-species fisheries with high levels of at-sea observers (strict enforcement). Chu [15] also suggested that inappropriate TAC levels and low levels of enforcement were likely reasons for stock declines in her study of ITQ fisheries.

If strict TAC levels and strict enforcement are effective in conserving fish stocks, then these methods should be more effectively promoted and incorporated into management schemes. The tremendous benefits of conservation and fishery re-building was studied extensively by Costello et al. [2]. They studied over 4500 fisheries and applied state-of-the-art bioeconomic models. Their findings indicate that in nearly every country in the world, fishery re-building and recovery would simultaneously drive increases in food production, fishery profits, and fish biomass (sustainability). A true win-win-win effect. They propose a suite of approaches using individual or communal access rights (rights-based approaches) that can align incentives across profit, food, and conservation goals so that few tradeoffs need to be made in selecting effective policy implementations.

If such benefits are possible worldwide without significant tradeoffs among important objectives, a reasonable question to ask is "Why haven't such policies been implemented more often?" In this

paper, we propose that rebuilding fisheries with ITQ require short-run sacrifices by fishers and the fishing community. Specifically, TAC setting policy may be influenced by cognitive limitations such as short-termism thinking and affective risk assessment.

## 2. Human Information Processing, Decision-Making, Bias and Affective Risk

Risk can be thought of and resolved in three fundamental ways: (1) affective risk, or risk as feelings, is when our fast, instinctive, intuitive and automatic reasoning provides a quick answer or affective reaction to risk; (2) risk as analysis uses slower cognitive processing involving logic, scientific reasoning, probability and evidence; and (3) risk as politics, when our first two modes of risk assessment are in conflict and must be resolved (see [22,23]). The first two modes of risk map to System 1 and System 2 of the Dual Process Model of Cognition (See Table 1).

**Table 1.** The Dual Process Model of Cognition: Comparing System 1 (Automatic) with System 2 (Analytical) [1].

| System 1, Experiential System (Automatic) | System 2 (Analytical) |
|---|---|
| Fast, automatic cognition | Slow, deliberate |
| Always available | Lazy, not always invoked |
| Uses associative memory, connections | Logical connections |
| Encodes reality in concrete images, metaphors, and narratives | Encode reality in abstract symbols, words, and numbers |
| Unconscious | Conscious evaluation of events |
| Holistic | Analytic |

[1] Adapted from [24,25].

System 1 cognitive processing is the automatic thinking that takes place behind the scenes in our subconscious [25,26]. System 1 is responsible for intuitive thinking, where we are able to make fast choices without long, slow, deliberative and conscious analysis. Heuristics, or mental short-cuts allow us to make choices and decisions relatively quickly. System 1 is a necessary capability for human decision-making because we cannot always afford slow deliberation, especially in a life-threatening situation. Experts in a particular domain often use heuristics as well. These heuristics are based on many years of experience where experts assemble and store information and are able to match cues in the environment to arrive at good decisions.

However, humans also arrive at decisions using System 1, even in domains where they have little expertise. Intuitive judgments are often based on emotion, feel or affect. In such cases, humans may use substitution, answering a similar question than what was posed, and answering based on feelings of liking or disliking. This is the use of the affect heuristic [25] (pp. 10–13).

Affective risk plays a dominant role in how people perceive risk. Many thoughts are encoded as images or perceptual representations. A lifetime of learning results in these images becoming marked by positive and negative feelings. The "affect pool" is the collection of these images that can then serve as cues for probability judgments [22]. When required judgments are complex or mental resources are limited, people make use of the availability heuristic [25] and may rely on mental shortcuts. In affective risk assessment, people base their judgments not only on what they *think* about the situation (System 2, analytical mode) but also on how they *feel* about it [22], essentially applying an *affect heuristic* to risk assessments.

There are certain biases in the experiential system (System 1) that can cause decision-makers to misjudge risk. Specifically, when outcomes must be evaluated that are visceral in nature, such as hunger, thirst, sexual desire, emotions, pain, and drug craving, they produce strong feelings in the present moment, but these feelings are difficult, if not impossible, to recall or anticipate [22,27]. Although currently experienced visceral factors have a disproportionate impact on behavior, delayed visceral factors tend to be ignored or severely under-weighted in decision-making [27] (p. 240). These cognitive tendencies can cause decision-makers to over value short-run outcomes over long-run goals.

## 2.1. TAC Setting, Sustainability and Human Decision-Making Bias

Setting a strict total allowable catch (TAC) limit is the essential management tool used in catch share and ITQ fisheries to manage stock levels and control long-run sustainability [21]. We consider sustainability from a broad perspective to include sustainable livelihoods or local community economic health, employment, and social equity to more global concerns such as maintaining high and stable harvests, and the ecological health of the fishery itself. Rebuilding fish stocks and ecosystem management can be considered primary objectives since other sustainability goals can be more easily managed from a strong and healthy fish stock and stable ecosystem. However, balancing these key objectives does create conflicts as higher employment levels and greater short-run community economic health depend on adequate catch levels.

In virtually all ITQ fishery management regimes, fishers are able to voice their concerns and exert political pressure on management. If long-run visceral impacts are under-weighted in decision-making, this could bias TAC levels higher. So, when evaluating the risk of catastrophic fishery collapse against a more immediate risk of financial hardship, fishers are naturally more concerned with and biased toward the present risk of financial failure and a loss of their way of life. When confronted with these pressures, fishery managers may be influenced.

These pressures from the community can be significant. Fishers make significant investments in gear and expect a certain level of productivity to remain financially viable. We know there is a strong cognitive tendency for decision-makers to over-value their possessions and to strongly prefer avoiding loss relative to a potential gain [28]. When faced with reduced TAC and catches, fishers naturally question the accuracy of the scientific assessments [29–32]. The increased technological capability of fishing gear and vessels adds to this problem because fishers are able to keep catch rates high despite information from fishery managers that stocks are low [33].

Meanwhile, fishery management must be concerned with precautions as stock rebuilding and long-run sustainability depend on restricting catch levels. The added complications to this picture include uncertain ecosystem health and uncertain feedback on stock status.

Roughgarden and Smith [34] maintain that to avoid collapse, a fishery needs to be maintained for ecological stability by keeping the stock level above that which produces the maximum sustainable yield (MSY) and harvested at less than the MSY. Maintaining a buffer in the stock level is referred to as "natural insurance."

Figure 1 shows a generalized graph of revenue across different levels of stock biomass based on the biomass dynamic model [35]. The region left of the MSY, labeled A, is dangerous because fishing pressure can lead to reinforcing feedback where harvests can drift above sustainable fishery output leading to lower stock biomass and less recruitment and less biomass growth. If feedback on fishery health is delayed or inaccurate, the fishery can spiral out of control and lead to a stock crash. In addition, natural environmental fluctuations can also send the fishery into collapse. Climate change adds to the uncertainty because of the many unknown impacts to ecosystems and subsequent adaptive responses [3]. The only sensible goal is to allow for natural insurance, create a buffer as a lower stock target, labeled B, and create a target stock zone, labeled C, in Figure 1. Unfortunately, there are short-run pressures in the system to increase catch rates and push the biomass toward lower and thus, riskier levels.

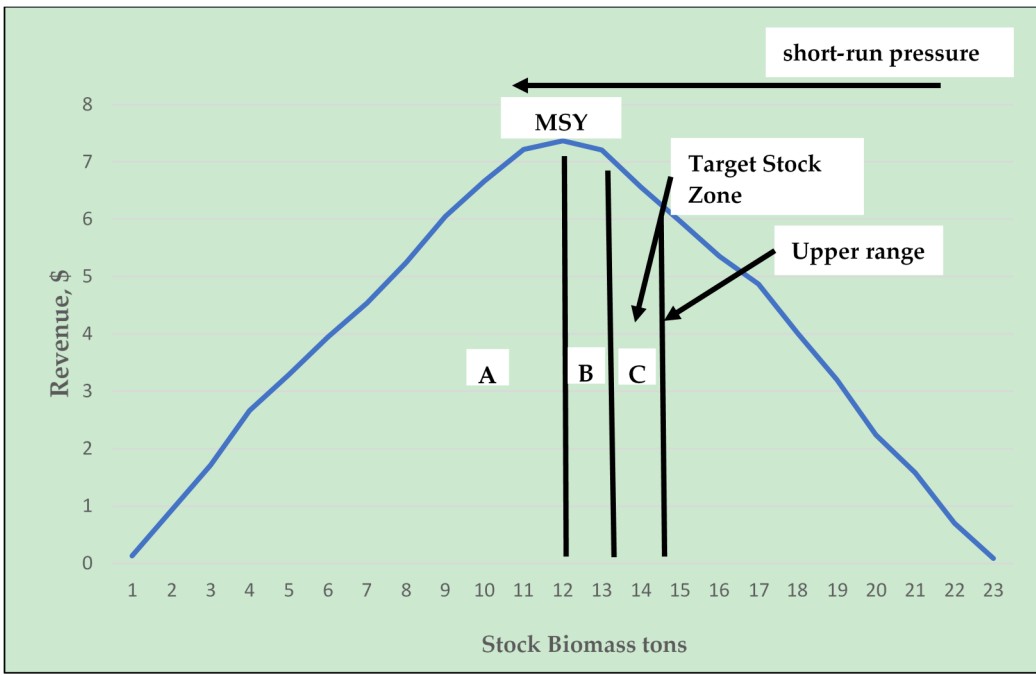

**Figure 1.** Risk, Biomass and Stock Target Zones. MSY = maximum sustainable yield.

*2.2. Short-Run Pressures and the People Component of the Fishery*

In ITQ fisheries, local fishers and quota holders have some economic and ecological incentives to seek lower TAC levels since the economic value of quota shares increases for well-managed fisheries [15,16]. However, there are several important issues that modify this incentive. First, ITQ shares do not create true property rights [36,37]. A quota share only creates a "right" to harvest a portion of the TAC but confers no real control over the resource itself. Any short-run sacrifice required to rebuild the fishery is born by the individual. However, the long-run benefits that result from stock rebuilding are shared by all fishery participants. So, in essence, the tragedy of the commons is not removed [38]. Fishers still benefit by cheating behavior such as high-grading, quota busting and misreporting catch.

Second, while fishers have economic incentive to see fish stocks well-managed in the long-run, they also face short-run financial pressure to stay profitable, survive and remain in the fishery. Lease fishers are sometimes in a difficult situation of having high, up-front capital investment costs and are thus, under higher financial stress [39]. Short-run financial pressure exerts a considerable amount of affective risk (i.e., fear) because the failure to achieve financial success is immediate and tangible. In contrast, the long-run viability of the fishery is further away in time and therefore, receives less affect in decision-makers' assessment of risk. In setting TAC levels, fishery managers face pressure from small-scale operators who struggle to remain financially viable. Figure 2 provides a comprehensive systems view of the important variables and pressures involved in TAC setting and fishery rebuilding.

Figure 2 shows the systems structure of an ITQ fishery as a causal loop diagram (CLD). The arrows between variables depict the direction of causal influence. There are two types of connections, either "+" or "−" polarity that show how a dependent variable will change. For example, a positive link means that if the cause variable increases, then the effect variable also increases above what it would otherwise have been. A positive link also indicates that if the cause variable *decreases*, then the effect variable also *decreases* below what it would otherwise have been. In essence, a positive causal link will *reinforce* the initial causal influence [8].

In contrast, a negative causal link will *balance* the initial causal influence. A negative link means that if the cause variable *increases*, then the effect variable will *decrease* below what it would otherwise have been (an opposite change). A negative link also indicates that if the cause variable *decreases*,

then the effect variable *increases* above what it would otherwise have been [8]. Double hash-marks indicate system delays and thus, causal links that may not be active until later. Balancing feedback loop B4, decline or rebuild, is bold-faced, indicating it is a dominant loop and of central importance. It is a balancing feedback loop with delays which can generate stock rebuilding, stock decline, or oscillation behavior, depending on the decisions related to TAC (quota).

In an ITQ fishery, a TAC is set as an overall limit to fishery catch. The individual transferable quota (ITQ) is then divided up among the participants in some fashion, as a percentage of the TAC. In this diagram, we use quota in a general sense, as information feedback that influences participant behavior. In the case of a fishery where stock rebuilding is necessary, we can trace the causal influences starting with a decreasing bio-economic sustainability (labeled "1. Bio-economic sustainability," Figure 2) relative to a target stock goal (see [8]). A decreasing bio-economic sustainability would then lead to an increase in pressure to lower quotas, then to a decrease in quotas, leading to a decrease in catch, then to an increase in the fish stock, leading finally to an increase in bio-economic sustainability. In essence, we can see fishery management's response to this problem or gap between the desired and actual state by tracing through this causal path.

Lowering the TAC to aid stock rebuilding means a lower catch. Lowering catch has the effect of increasing price which may mitigate reductions in revenue. Higher price allows less efficient fishers to stay in the market longer but eventually low TAC levels have the effect of pushing less efficient fishers out of the market and reducing fishing capacity. Not all causal loops in the model are dominant. For example, R6, Uneconomic stock levels, only becomes dominant when bio-economic sustainability is very low.

However, it is also important to note that management's actions are not performed in isolation. There are long-run impacts. Variables in the system are interdependent and changes in TAC lead to impacts in the fishery community. Thus, lower catches also lead to reduced revenue and lower levels of community fishers' economic health. Reductions in economic health can lead to various systemic cheating behaviors such as lobbying the political authority, employing direct pressure on management to raise quota, and quota busting (illegal catches above quota). The local community is policy-resistant because their short-term economic interests are impacted by fishery management.

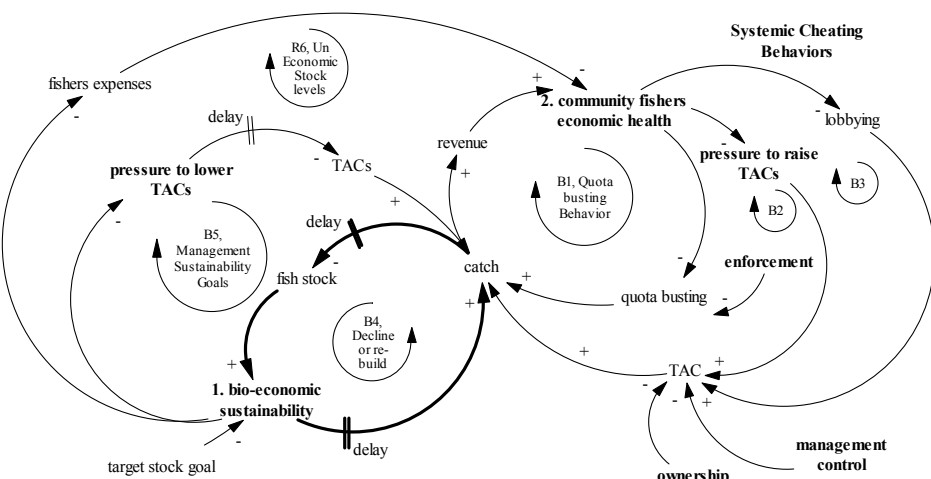

**Figure 2.** Policy Resistance, Diffused Responsibility and Ownership (adapted from [8]).

The system model in Figure 2 depicts an inherently unstable resource (the fish stock) that is controlled by a balancing feedback loop with delays (labeled "B4, Decline or rebuild"). Such a structure has the potential to exhibit oscillating behavior over time [40,41]. In essence, there are opposing forces attempting to control the resource through quota (or TAC) policy. Two exogenous variables are depicted: ownership and management control. High levels of ownership (or community stewardship) mean that quota setting will be lower and thus allow for stock rebuilding or long-run sustainable

management of the resource. High levels of community stewardship (ownership) are a feature of successful, decentralized common pool resource management systems [36]. A similar situation exists with the variable, management control, indicating that strong public management, resistant to outside lobbying efforts, will set lower quota. However, absent of these two controlling forces, the normal tendency in human behavior is to seek short-term (financial) advantage.

In addition to this already unstable system of managing the fish stock, is the additional complication of the separation of fishing activity from share ownership. In ITQ fisheries, share owners often lease shares to fishers. Under this arrangement, however, lease fishers (non-quota holders) have been empirically observed to prefer *more restrictive* TAC levels compared to the actual quota shareholders [42]. This appears to be counterintuitive since a standard rationale for ITQ systems is that quota holders have economic and ecological incentives for well-managed stocks, as quota owners can benefit from long-run rebuilding efforts.

### 2.3. Psychological Ownership as Distinct from Economic Ownership

We offer several potential explanations for lease fishers preferring more restrictive quota than ITQ owners. First, those who actually live and work in the local community, lease fishers, may develop feelings of psychological ownership of the resource. Psychological ownership is defined as the state in which individuals feel as though the target of ownership belongs to them [43]. Individuals can develop feelings of ownership for a variety of objects, material and immaterial in nature, including land and resources [43,44]. Van Dyne and Pierce [45] note that there are clinically-based observations suggesting that responsibility, caring, stewardship, and acts of citizenship are enhanced when individuals experience feelings of ownership toward the target object. This is consistent with the experience of other publicly owned natural resource areas, such as natural forests, where stewardship of the resource improves with the granting of private access rights [16,46].

Second, ITQ holders who lease their shares to others but do not fish may have altered attitudes as well. Jentoft et al. [47] and Schreiber [48] caution that implementing private rights, with their focus on economic efficiency, may shift attitudes and values in ways that encourage more individualistic, income-maximizing behavior. There are numerous empirical studies that support the idea that money changes people's motivations and promotes more self-sufficiency, independence and reduced helpfulness toward others [49]; (see [50] for a series of empirical experiments on money and its motivating effect). Non-local ITQ owners have profit incentive but lack strong ties to communities. Without physical or concrete ties to the fishing community, non-local owners do not have the requisite contacts necessary to develop feelings of psychological ownership. In essence, non-local ITQ holders who lease shares to others will have a strong profit incentive, yet have little or no opportunity to acquire a stewardship ethic through psychological ownership of the resource. So, non-local ITQ holders benefit from higher TAC because: (1) they capture profit through leases and higher revenues; and (2) quota owners looking to exit the fishery have a short-run incentive to view the condition of the resource in a positive light. Since a higher TAC sends a positive signal to the market, those looking to exit and sell quota have incentives for higher TAC regardless of long-run consequences [42].

In contrast, local fishers report feelings of enjoyment through participation and involvement in fishing activity itself [42]. These non-economic reasons provide a rationale to become stewards of the resource and to care for it over the long-term.

## 3. Fishery Management, TAC Setting and Risk

Fishery management is primarily concerned with scientific assessment and balancing objectives, including local community employment, fishery productivity or output, and long-run sustainability of the fishery. In co-management fisheries, numerous stakeholders provide input to decision-making. However, balancing objectives in fisheries is difficult because of the conflicting nature of the objectives. So, keeping employment levels and fishery output high implies higher catch levels that might imperil long-run sustainability.

The win-win-win benefits of high employment, harvest and profitability, and long-run sustainability can only be achieved by moving to and managing stocks in Zone C of Figure 1. Attempts to enhance community employment prior to rebuilding can be counterproductive. Meanwhile, rebuilding stocks requires short-run sacrifice by the fishing community. This is where the human element is important to success and where fishers, the local community, science and fishery managers must work together in a cooperative way. However, interactions between social and biological systems can create a policy trap [29]. Setting policy to achieve high employment may be successful in the short-run but it works against managing stocks in Zone C of Figure 1 and against maintaining long-run equilibrium stock levels. This is a natural feature of complex, non-linear feedback systems; that policy that works well in the short-run tends to backfire in the long-run [40,41]. Scientific assessments require input from both fishers and scientific survey data but both are imperfect and may contain bias and mis-reporting [29]. The data itself is subject to mis-interpretation, human error and bias [51]. Fishers may have difficulty accepting assessments and policy that indicate reductions in fishing pressure are needed when their own livelihoods are at risk [32]. Financial survival of fishers is a visceral effect and thus, carries more weight for the fishing community than a precautionary policy by fishery management that is potentially based on an erroneous stock assessment.

It is, thus, imperative to develop institutions and fishery management that incorporate the multiple perspectives and risk assessment inputs of the various stakeholders [52]. Good information feedback is necessary for management to assess stocks and set reliable TAC. Fishery managers, fishers and community representatives need to develop trust so that multiple viewpoints, values and science are openly shared and can be used to develop sound policy. Fortunately, stakeholder involvement may have the additional benefit of increasing feelings of psychological ownership through enhanced control [53]. Unfortunately, engaging fishers and the local community on the advantages of rebuilding is also entangled with communication problems that have been linked to a "two cultures" effect, institutional factors [32], or issues with multiple local interests [30].

Multiple perspectives related to social life mean that risk cannot be adequately addressed by one simple perspective [54], and risk cannot be separated from issues of power, justice and legitimacy [55]. Providing additional scientific or normative knowledge and analysis does not necessarily resolve conflicts [54,56]. For example, Kahan et al. [56] found that the amount of science knowledge had no effect on a subject's position on the issue of climate change. Solution aversion has been found to be a behavior where people are motivated to avoid information that does not fit their ideology or world view [57]. However, resolution of risk among affected parties and different social groups can be effective and may best be handled by dialogue informed by democratic values [54].

A necessary requirement for good co-management and participation is requiring broader and more equitable support on Regional Fishery Councils [58]. In essence, special interests can dominate these councils and undermine long-run sustainability goals at the expense of promoting more immediate financial goals. This would help to prevent 'regulatory capture' by industry while still advancing participatory co-management [58] (p. 204).

Finally, a potential alternative to the dilemma of short-term interests and communication problems is the use of the scientific approach applied to select ITQ fisheries in the form of pilot tests. Since rebuilding fisheries requires short-run sacrifices by local communities, government support could be used to help manage the transition to ecologically-managed, ITQ fisheries. In essence, government support in the form of transfer payments in the present can help the local community navigate the financial sacrifices necessary when rebuilding [7,8]. If innovative policy solutions could be successful as pilot tests, the approach used and the learning gained could be adapted and modified in other environments. Fisheries are complex, human, biological adaptive systems, so although it is unlikely or impossible that a "one size fits all" fishery management approach can be applied everywhere [37], pilot approaches can aid the learning process.

## 4. Summary and Conclusions

ITQ systems and other rights-based approaches combined with ecosystem-based management offer the potential to rebuild global fish stocks. Rebuilding stocks may result in win-win-win benefits of achieving increases in food production, fishery profits, and long-run sustainability [2].

However, rebuilding stocks in ITQ fisheries requires setting a strict total allowable catch (TAC). Unfortunately, the human component of the system often derails good TAC setting. Specifically, stock rebuilding requires short-run sacrifices by fishers and other stakeholders. Although local fishers may have feelings of attachment, psychological ownership and stewardship toward the fishery, this is no guarantee they will support restrictive TAC. Their stewardship ethic can change when fisheries are managed "at the edge." Financial pressures can cause an affective risk response and fishers may shift attitudes toward short-run financial survival and thus, higher TAC levels.

Meanwhile non-local owners who lack strong attachments to the community often have greater financial interests as well. Together, these special interests can exert pressure on management and proper TAC setting. Regional Fishery Councils can help to offset this influence by requiring good democratic participation by a wider group of affected parties.

However, for ITQ systems to work properly, fishers and the local community must be able to overcome short-run financial obstacles to benefit from longer-term rebuilding. In many cases, fishing communities may not be able to make the needed sacrifices to achieve a sustainable balance. Government transfer payments to local communities making such a transition could be helpful in moving these communities to a long-run, stable social-ecological system. Only government has the necessary assets and long-term time horizon needed to directly compensate the local community and ensure that they make a just transition to a sustainable state.

**Funding:** This research received no external funding.

**Acknowledgments:** Financial support was provided by the Richard J. Wehle School of Business, and the Marketing & Information Systems department, Canisius College.

**Conflicts of Interest:** The author declares no conflict of interest.

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
