# Peer review of "Individual Transferable Quotas (ITQ), Rebuilding Fisheries and Short-Termism: How Biased Reasoning Impacts Management"

_systems, doi:10.3390/systems8010007_

Round 1

Reviewer 1 Report

Two main concerns exist with the manuscript, one relates to risk and the other relates to fisheries social science.

the authors need to engage the extensive social science perspectives on risk that better inform this approach beyond this narrow deployment of psychological risk. In particular taking a look at the work by Ortwin Renn on Risk Governance makes sense. There is a broad range of risk approaches (social, cultural, systems) that should inform this particular view addressed in this article.

There is also a wealth of fisheries social science work that addresses the role of culture and attachment to fishing and the ramifications on fisheries conservation, especially in areas such as the Georges Bank and the New England cod fishery that shows how a range of cultures clash with scientific approaches to fisheries such as determining the TAC (Finlayson, Sinclair, etc see below).

Specific Comments

Need to define notions of sustainability, as currently used only infers population stability and not ecosystem nor true sustainability (social, economic, ecological). Sustainable fisheries are more than about MSY.

Missing whole host of risk research including other psychological dimensions from Slovic, to non-psychological risk approaches as covered in Renn's Risk Governance. A systems approach to risk might better address the situation given the nested systems of actors involved as well as the cultural approach (Douglas and others).

Figure 2 ignores price. The economic signal is price and profit so just because catch declines does not mean income declines. Ditto for costs. ITQs can reduce travel time and fuel required to access fishing stocks. Price has mixed effects sometimes positive and sometimes negative relationships. As prices increase there is an incentive to catch more fish.

There is a long history of psychological ownership leading to fishery ruin such as in New England and Canada groundfish. A strong attachment to a lifestyle can erode the logical risk perceptions depicted in the earlier figure. The references should include the long trend of anthropological and sociological work in this area. See all the books on the Canada fishery collapse.

Line 230. “Standard fishery economic theory predicts that when TAC levels increase, catch rates increase, total catch goes up, and eventually stock abundance decreases” This is only true when approaching MSY from the A zone. A recovering stock can increase TAC depending on the status of biomass in the stock. There are at least 2 points where harvest levels are the same, one in a declining fishery heading to ruin and the other in a thriving fishery well above MSY.

Line 238-240 “In this case, increasing the TAC means fishing effort will increase (in the long term), leading to greater costs for fishermen who lease quota. So, although revenue may increase for lease fishermen, profits may actually decrease due to higher fishing costs.” This is not necessarily true. TAC and shares are two different components. And again this ignores price. With ITQ there is no change in effort as measured in boats, gear, miles traveled, technology, etc. there are other explanations for increases in effort such as technology (even addressed in the Acheson et al. 2015 paper cited in the manuscript).

See Palmoares and Pauly recent paper on effort. https://www.ecologyandsociety.org/vol24/iss3/art31/

Maria L. D. Palomares, Daniel Pauly. On the creeping increase of vessels’ fishing power. Ecology and Society, 2019; 24 (3) DOI: 10.5751/ES-11136-240331

There are a range of constraints applied to a range of IFQ including limits on transferability, ownership, leasing, etc.

The section on “”Fishery Management and TAC Setting” (pg. 8 266-270)  ignores the appropriate set of risk governance schemes that govern fishery management and TAC setting (even their failures such as the industry capture evident in the Fishery Management Councils in the U.S.). These are more institutional and systems risks (eg. Luhman, Renn and others) and not psychological. The psychological might apply more to individual fishers, but not to the systems of fisheries management.

Don’t see sufficient evidence for the possible effectiveness of the recommendations “(1) Rules that restrict non-local owners from having share ownership; and (2) Lease agreements that require lease fishers’ representation on co-management committees in proportion to their catch shares.” (pg 8 279-281) 

Finlayson, A. C. (1994). Fishing for truth: A sociological analysis of northern cod stock assessments from 1977-1990. St. John's, Newfoundland, Canada: Memorial University of Newfoundland, Institute of Social and Economic Research.

Sinclair, P. R. (1996). Sustainable development in fisheries dependant regions? Reflections on Newfoundland Cod fisheries.Sociologia Ruralis, 36(2), 225-235.

Arnason, R., Hannesson, R. & Schrank, W. E. (2000). Costs of fisheries management: The case of Iceland, Norway and Newfoundland. Marine Policy, 24(3), 233-243.

Haedrich, R. L., & Hamilton, L. C. (2000). The fall and future of Newfoundland's cod fishery. Society & Natural Resources, 13(4), 359-372.

Palmer, C. T. & Sinclair, P. R. (1996). Perceptions of a fishery in crisis: Dragger skippers on the Gulf of St. Lawrence cod moratorium. Society & Natural Resources, 9(3), 267-279.

Finlayson, A. C., and B. J. McCay . 1998. Crossing the threshold of ecosystem resilience: the commercial extinction of northern cod. Pages 311–337 in F. Berkes and C. Folke, editors. Linking social and ecological systems: management practices and social mechanisms for building resilience. Cambridge University Press, Cambridge, UK.

DOUGLAS CLYDE WILSON (2003) Examining the Two Cultures Theory of Fisheries Knowledge: The Case of Bluefish Management, Society & Natural Resources, 16:6, 491-508, DOI: 10.1080/08941920309150   Okey, T. 2003 Membership of the eight Regional FisheryManagement Councils in the United States: are special interests over-represented? Mar. Pol.27, 193–206.   Renn, O. 2008. Risk governance: coping with uncertainty in a complex world   Renn, O. (1992). Concepts of risk: A classification. In S. Krimsky & D. Golding (Eds.), Social theories of risk (pp. 5379). Westport , CT : Praeger. https://elib.uni-stuttgart.de/handle/11682/7265
  Douglas, M. & WILDAVSKY, A., (1983) Risk and Culture: An Essay on the Selection of Technological andEnvironmental Dangers, p. 221 (London, University of California Pres   Tansey, James & Riordan,. (1999). Cultural theory and risk: A review. Health, Risk and Society. 1. 10.1080/13698579908407008. http://lchc.ucsd.edu/mca/Mail/xmcamail.2014-11.dir/pdfRk8rlpiptu.pdf

Reviewer 2 Report

This is a very well-written paper. I think it has the potential to be a great addition to our field. However, I have three main concerns with the manuscript in its current state: (1) in its current state, the manuscript appears to be a summary of the field of rights-based fisheries, (2) there is a lack of seminal papers on rights-based fisheries, proper rights, and cognitive psychology that was not referenced, and (3) the cognitive psychological lens needs to be more prominent throughout and it needs to be more apparent how viewing RBM from this angle will aid management approaches. Due to these reasons, I have provided some suggestions for consideration. I have provided more detail comments in the attachment. 

Round 2

Reviewer 1 Report

There is growing recognition that contests the notion stated here that sustainable development requires "growth" in any form whether it is profits, populations, or other components. Development (improvement in conditions) exists independent of growth. Growth can actually constrain development such as with rising inequalities in income or conditions. Equating growth with sustainable development is a faded concept, though still the dominant paradigm. The last statement is in fact the goal of sustainable development "maintain ecosystems at long-run sustainable levels may not be compatible with supporting human populations  that depend on them." See following references:

https://sustainabledevelopment.un.org/content/documents/617BhutanReport_WEB_F.pdf

http://www.clubmadrid.org/wp-content/uploads/2017/11/Shared_Societies-Report-13.pdf

https://www.weforum.org/agenda/2019/12/davos-manifesto-2020-the-universal-purpose-of-a-company-in-the-fourth-industrial-revolution/

https://www.kateraworth.com/(explicitly focuses on the just and safe space for humanity by meeting the social foundations for all, without exceeding planetary boundaries)

pp 12-13

433>"Extending the idea of sustainability to include the notion of sustainable development is a basic underlying problem with trying to manage tightly coupled socio-economic ecological systems. Sustainable development contains the idea that we can grow and sustain profits and communities
without causing harm to our ecosystems upon which we depend. Alternatively, attempts to maintain ecosystems at long-run sustainable levels may not be compatible with supporting human populations  that depend on them (pp. 224-225) [60]."

438>

Reviewer 2 Report

In my opinion, the author has made satisfactory progress in addressing my initial comments on this manuscript. In its present revised form, this manuscript is clearer and offers a more streamlined message. I do believe that the summary and conclusions need to be stronger. In its current state, it does not adequately present a true reflection of the manuscript. Please see below two minor requested revisions:

Line 188-189: delete “Climate change adds to the uncertainty” or provide a one-sentence description of your explanation.

Line 306-312: delete: “however, incentives are strongly…” this statement is repetitive and has been mentioned already in the previous sections of the manuscript.

With these suggested revisions, this manuscript is now acceptable for publication. 
